# SERS Signature of SARS-CoV-2 in Saliva and Nasopharyngeal Swabs: Towards Perspective COVID-19 Point-of-Care Diagnostics

**DOI:** 10.3390/ijms24119706

**Published:** 2023-06-03

**Authors:** Sylwia M. Berus, Ariadna B. Nowicka, Julia Wieruszewska, Krzysztof Niciński, Aneta A. Kowalska, Tomasz R. Szymborski, Izabela Dróżdż, Maciej Borowiec, Jacek Waluk, Agnieszka Kamińska

**Affiliations:** 1Institute of Physical Chemistry, Polish Academy of Sciences, Kasprzaka 44/52, 01-224 Warsaw, Poland; sberus@ichf.edu.pl (S.M.B.); anowicka@ichf.edu.pl (A.B.N.); jwieruszewska@ichf.edu.pl (J.W.); knicinski@ichf.edu.pl (K.N.); tszymborski@ichf.edu.pl (T.R.S.); jwaluk@ichf.edu.pl (J.W.); akaminska@ichf.edu.pl (A.K.); 2Department of Clinical Genetics, Medical University of Łódź, Pomorska 251, 92-213 Łódź, Poland; izabela.drozdz@umed.lodz.pl (I.D.); maciej.borowiec@umed.lodz.pl (M.B.); 3Faculty of Mathematics and Science, Cardinal Stefan Wyszyński University, Dewajtis 5, 01-815 Warsaw, Poland

**Keywords:** SARS-CoV-2, saliva, nasopharyngeal swabs, surface-enhanced Raman spectroscopy, chemometric analysis

## Abstract

In this study, the intrinsic surface-enhanced Raman spectroscopy (SERS)-based approach coupled with chemometric analysis was adopted to establish the biochemical fingerprint of SARS-CoV-2 infected human fluids: saliva and nasopharyngeal swabs. The numerical methods, partial least squares discriminant analysis (PLS-DA) and support vector machine classification (SVMC), facilitated the spectroscopic identification of the viral-specific molecules, molecular changes, and distinct physiological signatures of pathetically altered fluids. Next, we developed the reliable classification model for fast identification and differentiation of negative CoV(−) and positive CoV(+) groups. The PLS-DA calibration model was described by a great statistical value—RMSEC and RMSECV below 0.3 and R^2^_cal_ at the level of ~0.7 for both type of body fluids. The calculated diagnostic parameters for SVMC and PLS-DA at the stage of preparation of calibration model and classification of external samples simulating real diagnostic conditions evinced high accuracy, sensitivity, and specificity for saliva specimens. Here, we outlined the significant role of neopterin as the biomarker in the prediction of COVID-19 infection from nasopharyngeal swab. We also observed the increased content of nucleic acids of DNA/RNA and proteins such as ferritin as well as specific immunoglobulins. The developed SERS for SARS-CoV-2 approach allows: (i) fast, simple and non-invasive collection of analyzed specimens; (ii) fast response with the time of analysis below 15 min, and (iii) sensitive and reliable SERS-based screening of COVID-19 disease.

## 1. Introduction

The pandemic of coronavirus disease 2019 (COVID-19), caused by severe acute respiratory syndrome coronavirus 2 (SARS-CoV-2), burdens the economy and the healthcare around the globe. This disease was first reported in December 2019 in Wuhan, China, and resulted in more than 6,000,000 deaths (as of May 2023) [1].

The fast human-to-human spread of COVID-19 infection across the world is related to (i) its highly infectious properties, (ii) easy transmission through respiratory droplets (saliva, nasal discharge) [2] (iii) direct contact routes via oral, nasal, eye mucous membrane, (iv) large number of asymptomatic cases [3,4,5,6], and (v) high mutation rates of viral RNA [7,8,9]. Nowadays, mainly symptomatic patients are tested and we still do not catch a large number of people who pass the infection asymptomatically and may infect others. Hatching a network of contacts is crucial with the British variant of SARS-CoV-2, which is much more contagious [10,11,12]. Developing the new testing strategies are central to gather information about the presence and propagation of SARS-CoV-2 in the population, understand the COVID-19 disease at different stages, and to monitor the effectiveness of vaccinations in order to estimate the prevalence of immunity (population surveillance) [13,14,15,16]. Typically, two types of tests were used for detection of COVID-19: (1) molecular diagnostic test that detects the presence of the virus (viral genetic material (RNA) in a patient sample) and (2) serologic test that detects the immune response to the virus (antibodies against various SARS-CoV-2 proteins (including the spike protein, nucleocapsid protein, and receptor-binding domain) [17,18,19,20,21,22,23]. The first and most common test for detection of SARS-CoV-2 infection is based on real time reverse transcription polymerase chain reaction (RT-PCR) on nasopharyngeal swabs [24]. Although this technique possesses advantages, including very high sensitivity and specificity, the false negative and false positive results are also possible [25,26,27,28]. According to WHO, a number of factors could lead to an incorrect result during RT-PCR analysis, caused by the technical reasons inherent in the test, virus mutation or PCR inhibition [29]. Moreover, this method is time-consuming (the results are received after at least few hours and, in some cases, more than a 24 h period), requires the purchase of expensive reagents, and is a labor-intensive procedure. In consequence, the number of tests performed per day is strongly limited. The serologic methods (e.g., chemiluminescent and enzyme-linked immunosorbent assays, ELISA) are based on the detection of antibodies engaged against the virus. As the knowledge on SARS-CoV-2 antibody kinetics is limited, the optimization of immune-response based tests and proper interpretation of readings is still challenging. Therefore, the immunological methods are not applicable for early COVID-19 diagnosis.

New methods enabling fast and reliable detection of SARS-CoV-2 are extremely desired. Surface-enhanced Raman spectroscopy (SERS) can be successfully used for such detection, as it is very sensitive technique which can be performed in a label-free manner. The SERS technique enhances the Raman signal when analyzed molecules are close to metallic nanostructure, roughened surface or nanoparticles, usually silver, gold, and copper [30]. According to scientific reports, there are two phenomena contributing to total enhancement of normal Raman signal: (i) electromagnetic connected with the excitation of localized surface plasmons; (ii) chemicals which arise due to charge transfer between analyzed molecules and the surface [31,32,33]. High enhancement (typically 10^6^–10^8^) make the SERS a promising method for single molecule detection with very high specificity and sensitivity which can be applied in pharmacology, chemistry, medicine, and for imaging biological systems, e.g., viruses, yeasts, bacteria, and cancer cells [34,35,36,37,38]. Interestingly, SERS can be used to study RNA, e.g., to identify and quantify AS genes [39], to detect four different RNA species from plant [40] and also to detect the effect of T-DNA insertion on mRNA with high sensitivity and precision [41]. The results of the SERS measurements can be obtained in a few minutes and no additional reagents are needed. Currently, the complex spectroscopic responses from biological specimens, especially from clinical samples, are analyzed and resolved via chemometric methods and artificial intelligence. Such features, together with the highly sensitive and selective nanoplasmonic SERS substrates are very advantageous in COVID-19 detection. Carlomagno et al. demonstrated that saliva can be used for discrimination between patients with current and past infection of SARS-CoV-2 and the uninfected ones [42]. The similar studies on saliva samples was performed by Ember et al., who obtained the specificity and sensitivity at the level of 84% and 64%, respectively [43]. Karunakaran et al., using SVM, obtained 95% accuracy in the differentiation between patients with and without COVID-19 [44]. The SERS spectra of nasopharyngeal swabs (NHS) was also analyzed by Yang et al., but no significant differences between infected and noninfected group of NHS samples was observed [45].

In laboratory practice, to evaluate patients with suspected respiratory infection caused by viruses and bacteria, mainly RT-PCR and immune-based recognition assays are applied. In the present study, we developed a biosensing platform based on femtosecond laser-modified silicon integrated with a small, portable Raman spectrometer for fast, label-free, cheap and reliable diagnosis of COVID-19 from nasopharyngeal swab and saliva.

The clinical application of SERS-based biosensor on solid nanoplasmonic platform for COVID-19 diagnosis was not yet presented. Considering the complexity of spectral images of tested biological specimens additionally enriched by the expression of the potential COVID-19 biomarkers such as ACE2, immunoglobulins G, M, and A, microRNAs, adenosine deaminase [46], multivariate analysis approaches were applied to record SERS datasets. We utilized supervised methods, such as PLS-DA, PCA-LDA, and SVMC, to screen important spectroscopic features reflecting the main biochemical differences between infected and uninfected samples. In addition, these methods offer the classification of new samples based on a previously created and optimized calibration model. Therefore, it becomes possible to identify the origin of a sample and determine its class membership (CoV(+) or CoV(−)) in a rapid and automated manner. This type of solution would improve point-of-care diagnostics [47,48].

The clinical sensitivity of SARS-CoV-2 detection varies in different types of clinical specimens [49] and, thus, it is very important not only to develop a novel detection approach, but also to examine its diagnostic accuracy for different clinically acceptable testing specimens. We performed a perspective investigations to: (i) study the changes in biochemical composition of saliva and nasopharyngeal swabs associated with viral infection; (ii) indicate the SERS-biomarkers for COVID-19 immunopathology controlling and detection; (iii) create the classification model (with validation stage) to develop the automated procedure of classification of patient’s samples into two groups CoV(+) and CoV(−) using chemometric analysis, and (iv) calculate the sensitivity of SARS-CoV-2 detection from both saliva and nasopharyngeal specimens.

## 2. Results and Discussion

For the SERS studies, the saliva and nasopharyngeal swabs samples were included from patients being tested for SARS-CoV-2 by the qRT-PCR method, according to the Department of Clinical Genetics, Medical University of Łódź guidelines. Figure 1 illustrates the subsequent operation steps for the SERS-based COVID-19 diagnosis.

### 2.1. Salivary SERS Fingerprint of COVID-19

We performed SERS investigations of saliva samples taken from 149 human donors (72 participants tested positive and 77 participants tested negative by the PCR method) on Si/Ag SERS platforms. The platforms were recently developed and optimized in the context of biological and medical samples in our group [50]. The averaged SERS spectra with the variation of the saliva spectra for each of the two groups CoV(+) and CoV(−) are presented as standard deviation SD (ribbon-like plots) in Figure 1A. The differences between the spectra of different participants and points are among 15–39% for both CoV(+) and CoV(−) data. The observed SD are related to the patient-specific features of the SERS spectra within the same group and result from some individual factors, i.e., the stage of disease or the organism’s response to the disease, and may manifest in different ways (the spectrum may be devoid of some bands). In order to more deeply analyze the observed spectral variations, as well as the reproducibility of spectra, Appendix A presents averaged spectra for saliva taken from five patients infected and five patients non-infected with COVID-19 (representing these two considered groups CoV(+) and CoV(−)). All recorded 15 spectra for each sample (marked as red for CoV(+), and green for CoV(−)) were superimposed along with the corresponding averaged spectrum (marked as black). All the spectra for each patient within the same group revealed similar bands, whereas the differences referred to the relative intensities between them. For negative saliva subject CoV(−), the most characteristics bands were observed at 691, 724, 853, 878, 1002, 1047, 1128, 1270, 1325, 1452, 1590, 1690, and 1792 cm^−1^. Among the most intense bands: (i) 724 cm^−1^ corresponds to O–O stretching vibration in oxygenated proteins, glycoproteins such as mucin, and to ring breathing mode of tryptophan, (ii) 1325 cm^−1^ is characteristic to amide III band in proteins and/or DNA (G and A) mode, (iii) 1452 cm^−1^ is assigned to the C-H stretching of glycoproteins including mucin, (iv) 1585 cm^−1^ is associated with ring and C=C vibrations in tyrosine and phenylalanine.

The spectral response for CoV(+) saliva revealed similar features located at the same Raman frequencies, yet some differences associated with their relative intensities can be noted. Spectral changes may reflect the biochemical composition and provide information about interactions between components. The intensity ratio of the characteristic tyrosine doubled at 828 and 853 cm^−1^, indicating the nature of tyrosine residues [51] is changing upon SARS-CoV-2 infection. In the SERS spectra of CoV(−) saliva samples, the intensity ratio of 853/828 cm^−1^ equaled 4.8. The intensity of the 853 cm^−1^ band significantly decreased for the CoV(+), bringing the intensity ratio 853/828 cm^−1^ down to a value of 0.8, which could be explained by some specific interaction between tyrosine residues that now are hydrogen-bonded acceptors with viral proteins (spike glycoproteins) or other expressed molecules (ACE2 receptor) [52] and immunity proteins (IgA, IgM, IgG) [53].

The SERS fingerprint of SARS-CoV-2-infected saliva subject (CoV+) was characterized by the bands at 654, 720, 1320, 1443 cm^−1^, which can be assigned to some specific oscillations in methionine (e.g., C-S stretching and CH_3_, CH_2_ deformation) or methionine adenosyl transferase (enzyme which converts methionine to S-adenosylmethionine) [54]. The intensity ratio of the individual bands to the band at 1002 cm^−1^ (I_654_/I_1002_, I_720_/I_1002_, I_1320_/I_1002_, I_1445_/I_1002_) was always higher for CoV(+) samples (Appendix A), yielding information about elevated production of methionine during infection. Such a conclusion can be supported by two phenomena that were discovered so far. First, S-adenosylmethionine, in the presence of nonstructural proteins (e.g., nsp10, nsp14, and nsp16) of SARS-CoV-2 was involved as a methyl donor in the process of viral RNA cap methylation (transferring a methyl group to the viral genome). This process was essential for virus replication, translation, and continued survival in host cells and the efficiency of S-adenosylmethionine synthesis is strictly correlated with the level of methionine [55]. If the concentration of methionine is too low, the viral methyltransferases reaction will be blocked. The reduced ability of the virus to proper replication, ultimately impinges on degradation of viral genomes. Second, during SARS-CoV-2 infection, T-cells and macrophages are over-activated, resulting in a huge increase in the level of cytokines (IL-1β, IL-6, TNF-α) and proteins associated with the macrophage. Upon this activation the T-cells also have an increased requirement for methionine [56]. The presence of T cells itself can be manifested by the 1094, 1325, 1372, 1452, and 1690 cm^−1^ bands, where their increased intensity ratios, with respect to the band at 1002 cm^−1^ for CoV(+) saliva, only prove the mentioned overactivation.

Ferritin is a protein that stores iron and releases it during cell proliferation or metabolic renewal. The study of ferritin levels is significant from the point of view of its deficiency as well as excess which implicates various diseases (e.g., anemia) [57]. Recent studies showed that salivary ferritin levels can rise significantly during the course of COVID-19 infection [58]. Immunoglobulins (e.g., IgG) are produced by cells in organisms of immune system in a response to some external agent (bacteria, viruses, protozoans). These proteins, having the ability to recognize and specific bind to their antigens, lead to the inactivation of microorganisms [59]. In these studies, the observed intensified bands in the region 1200–1300 cm^−1^ and the bands at 1325 cm^−1^, 1450 cm^−1^, 1690 cm^−1^ coming from amide III, amide I of proteins may reflect the increased level of ferritin [60] as well as specific immunoglobulins [61] in CoV(+) saliva.

In the SERS technique, bands can be assigned in an approximate way, as some of them have multiple origins, e.g., the previously mentioned bands at 1094, 1242, 1325 cm^−1^ can additionally arise because of individual (PO_2_^−^, phosphodiester group, and purine bases of nucleic acids) highly associated with DNA/RNA bases. Their increased intensity for CoV(+) saliva can be explained by the multiplication of genetic material during infection. The proposed band assignments of the SERS spectra are presented in Table 1 and the SERS spectra of the most prominent amino acids and peptides are shown in Appendix A.

#### 2.1.1. The Classification and Prediction Methods for Diagnosis COVID-19 in Saliva Samples

Spectral data are particularly difficult to analyze empirically, especially when there are subtle differences between them. Thus, chemometric methods can extract that part of the spectral information which differentiates the analyzed groups.

In the present work, the supervised and classification methods such as PLS-DA, PCA-LDA, SVMC were used to (i) create and develop a calibration model, (ii) check its classification abilities (validation stage), and (iii) extract the spectral information that is indicative of COVID-19 disease. The analyzed biological systems (saliva and nasopharyngeal swabs) are characterized by enormous biochemical complexity and variability from patient to patient. In addition, the SERS substrates have so-called ‘hot-spots’ that are responsible for generating the SERS signal. Therefore, different qualitative and quantitative distribution of the biomolecules at these sites can alter the SERS signal in several ways. Consequently, the chemometric analysis (creation of the calibration model, the external validation) was performed and considered at the level of a single spectrum. For each mentioned method (PLS-DA, PCA-LDA, SVMC), the calibration model was built from 72 CoV(+) and 77 CoV(−) saliva samples, and ten CoV(+) and ten CoV(−) saliva samples were then used to check classification abilities and mimic real diagnostic conditions. Each sample was measured in 15-points mapping mode, the total number of 2220 and 300 spectra were used at the stage of calibration and validation, respectively. Based on the results, important parameters (accuracy, sensitivity, specificity) that describe every diagnosis method were also determined according to the following formulas:(1)$$Accuracy=\frac{TP+TN}{TP+TN+FP+FN}\cdot 100\%,$$
(2)$$Sensitivity=\frac{TP}{TP+FN}\cdot 100\%,$$
(3)$$Specificity=\frac{TN}{FP+TN}\cdot 100\%,$$
where: TP—True positive, FP—false positive, TN—true negative, FN—false negative.

#### 2.1.2. Prediction by Means of PLS-DA Analysis

In the PLS-DA analysis, all spectral information in the range of 500–1900 cm^−1^ created the X matrix (explanatory variables), while the corresponding Y matrix values (e.g., ‘1’ stands for CoV(+) and ‘0’ stands for CoV(−)) are response variables. As the threshold line that effectively separated these two classes equaled 0.5, the predictive values >0.5 mean that the sample belonged to the CoV(+) group, while the value <0.5 means that the sample belonged to the CoV(−) group. The calibration model was established with the use of the non-linear iterative partial least square (NIPALS) algorithm and random cross validation (number of segments equal to 10). All the statistical parameters that describe the quality of calculations at different stages are gathered in Appendix A. Given the enormous biochemical variation in clinical samples resulting from variety of individual factors, the model was characterized by a relatively high R^2^ and low RMSE values for calibration and cross validation. The calculated number of latent variables indicates that 11 of them can sufficiently explain the whole spectral variance within the analyzed trained data. The score plots (Figure 1B) in 2D (above) and 3D dimension (below) for the three most influential factors show that two classes of CoV(−) and CoV(+) saliva samples were largely separated from each other to form clusters. The clusters resulted from the differences between the samples—from the same person it formed smaller clusters, later everything consists of a larger one, determining CoV(+) and CoV(−). All these variations in Raman spectra, as was already mentioned, were due to specific immune responses to this particular disorder. On the other hand, overlapping of these classes to some extent can be elucidated by the inaccuracy of PCR technique (false negative or false positive results). The first latent variable (LV1) explained 40% of the variance in the block Y with 22% of the spectral data (X matrix) while the LV2 explained 5% with 21% of the data within X matrix. The weighted regression coefficients that summarized the relationship between predictors and responses calculated for the approximation of three components (F-3, CoV(−) response) is presented in Figure 1C. This plot revealed variables corresponding to the main Raman bands observed on the averaged spectra (Figure 1A). A positive value of coefficient shows a positive link with the response, and a negative coefficient shows a negative link. Hence, saliva variables such as: 691, 858, 1002, 1047, 1132, 1605, 1785 cm^−1^ had positive coefficients and 654, 721, 956, 1094, 1238, 1568, 1698, cm^−1^ had a negative coefficient. The variables that had the largest value of regression coefficient play a significant role in a regression model and for saliva, it was 721 cm^−1^ with 0.05 weight and also 691, 1605 cm^−1^ (0.03 weight), and 1094 cm^−1^ (0.02 weight).

The loading plot that can be calculated for particular association and along the specific factor recognizes variables that affect the separation between the analyzed classes to a varying degree—those with the highest weight being the most influential.

For the saliva dataset, all the eleven factors presented the contribution of different variables with different weights (Appendix A). Factor-1 showed that most information indicates that the bands with positive values located at 690, 854, 1602 cm^−1^ were correlated with the samples on the positive sides of the score plot being recognized as CoV(−) and these with negative values 1452, 1695 cm^−1^ were characteristic for CoV(+) samples. For Factor-2, the most significant variable was 690 cm^−1^ and 1458 cm^−1^, and for Factor-3, 721 cm^−1^ and 1602 cm^−1^. The rest loading plots revealed the importance of different bands but at the diminished level of explained information (see Appendix A).

As was mentioned above, in the step of calibration, we used 149 saliva samples—77 CoV(−) and 72 CoV(+) and then 20 external data sets—ten CoV(+) and ten CoV(−) were used to test the predictive abilities of such prepared calibration model. All the diagnostic parameters were calculated at calibration as well as validation stage and referred to the single patient. In order to present the results of the analysis in an explicit way, the calculated predictive values and standard deviations for each spectrum were averaged within the sample (validation samples in Appendix A). Hence, at the stage of calibration, as most samples were correctly classified, the parameters were as follows: accuracy 93%, sensitivity 97%, and specificity 90%. For most of samples, the predictive values were close to 1 or 0 depending on the origin of the sample (Appendix A). For the three samples stated as CoV(−), predictive values exceeded the threshold line—0.5 meaning that they are more likely to be CoV(+) rather than CoV(−). Only one CoV(+) sample (number 4) had an incorrect predictive value. Therefore, the parameters were as follows: accuracy 80%, sensitivity 90%, and specificity 70% (Table 2). For the majority of samples, the values of standard deviation (STD) were in the range 0.16–0.34.

#### 2.1.3. Classification Results of PCA-LDA and SVMC Analysis

The PCA-LDA analysis was preceded by the PCA calculation to reduce the dimensionality of spectral data. The first 12 principal components explaining 98% of spectral variance were used as input data for future LDA investigations. Appendix A demonstrates the PCA-LDA discrimination plot for the CoV(+) and CoV(−) classes of the calibrated model. All samples are displayed and color-coded by class where red dots and blue dots represents CoV(+) and CoV(−) samples, respectively. These two exes relate to both groups and samples that belong to a given class were found to lie close to the value of zero of a given class. The quadratic algorithm that best described the spectral differences among trained data gave an 89% accuracy at the level of single spectrum. The determination of diagnostic parameters was carried out in the stage of calibration and in the stage of validation (with a special care that they refer to a single patient, not a single spectrum). The decision on assigning an appropriate sample to a particular class was made based on majority of spectra assigned to the particular class, so according to the formula X = n + 1 (where n is the number of spectra belonging to the particular class). Consequently, the created PCA-LDA calibration model provided an accuracy of 88%, a sensitivity of 97%, and a specificity of 79%, with respect to the total of 149 saliva samples (Table 2). The validation with the use of external samples was aimed at determining how such a prepared model would work in real diagnostic conditions. In order to deviate from suggestions and provide conditions that were as close to the real ones as possible, all information about the origin of a sample was delivered after measurements. The results of classification at the level of a single spectrum and the final response are summarized in Appendix A. PCA-LDA classified ten out of ten CoV(+) and six out of ten CoV(−) saliva samples, giving the accuracy 80%, sensitivity 100%, and specificity 60%.

SVMC calculations were performed with the use of Polynomial Kernel function with degree of three and cross validation with ten segments. The parameters were selected in such a way as to ensure the best accuracy of the calibrated model and, thus, to prevent overfitting. Hence, under the conditions of C = 0.1, γ = 0.1, the model provided 100% of the training and validation accuracy calculated at the level of a single spectrum. Therefore, the diagnostic parameters for the analysis of a single patient were: 100% sensitivity, 100% specificity, and 100% accuracy. At the stage of external validation, SVMC successfully recognized more samples than PLS-DA and only two of ten CoV(−) samples were misclassified (accuracy 90%, sensitivity 100%, and specificity 80%).

The obtained results demonstrate that SERS, supported by all considered methods, is promising and can be successfully applied for discrimination between CoV(+) and CoV(−) saliva. Taking into consideration the values of diagnostic parameters (see Table 2 for comparison), SVMC seems to be the best suitable method, as it provides the highest sensitivity, specificity and accuracy at the stage of calibration and validation. While creating calibration model, PLS-DA also correctly recognized a huge number of samples that is indicated by values accuracy 93%, sensitivity 97%, specificity 90%. Comparing PCA-LDA vs. PLS-DA at the stage of classification external samples, PCA-LDA can provide higher sensitivity and lower specificity while the accuracy of both methods remains the same. Therefore, based on the presented data, SVMC is more favorable model for saliva CoV(+) and CoV(−) discrimination.

### 2.2. Nasopharyngeal SERS Fingerprint of COVID-19

A nasopharyngeal swab is used to diagnose SARS-CoV-2 infection using mainly the PCR method [73]. The nasopharyngeal swab is a complex mixture of lipids (phosphatidylcholine), proteins (mostly surfactant proteins) that derive mainly from submucous glands, goblet cells, and transepithelial ion and water transport. Cytokines that play a significant role in airway disease can also make a contribution. The presence of some nonpathogenic and pathogenic bacteria (*S. pneumoniae*, *H. influenza*, or *M. catarrhalis*) were also proved. Several different upper and lower respiratory tract viruses (e.g., rhinovirus, adenovirus, influenza, respiratory syncytial virus (RSV), human parainfluenza viruses (HPIV)) can be detected through nasopharyngeal swabs [74,75]. However, the nasopharyngeal swab is less of a chemically complex matrix than saliva; therefore, its applicability as tasting specimen in label-free configuration of SERS analysis is advisable. The protein amount in nasopharyngeal swab is lower in comparison to saliva and blood. We expect that quantitative and qualitative changes of various proteins in naso-oropharyngeal samples of SARS-CoV-2-infected patients would likely be related with viral associated proteins that manage the host antiviral protection system. Clinical and experimental studies indicated an accumulation of saturated and unsaturated fatty acids and phospholipids and increased concentration of immune markers such as the IL-6 and CXCL10 cytokines [76,77].

Figure 2A presents the nasopharyngeal swab SERS spectra together with SD analysis of the whole biochemical pattern of two considered experimental groups: CoV(+) and CoV(−). We detected a significant spectroscopic pattern in the nasopharyngeal swab of SARS-CoV-2-positive participants in comparison to SARS-CoV-2-negative participants. In both CoV(+) and CoV(−), the strong SERS bands located at 724, 1002, 1045, 1330, 1452, 1590, and 1680 cm^−1^ can be consistently observed. We also observed distinct spectral differences in the 680–950 cm^−1^ regions. In the SERS spectra of CoV(+) nasopharyngeal samples, a new band appeared at 688 cm^−1^ that can be assigned to neopterin as a marker band for disease severity; one also observed an increase in the intensity of the band at 925 cm^−1^ assigned to carboxylates and compounds with proline rings [78,79]. Neopterin is the pteridine derivative containing 2-amino, 4-oxo, pyrimidyno-pyrazino ring produced by human monocytes and macrophages after activation by interferon-gamma (IFN-γ). Neopterin is known and utilized as the inflammatory marker of the cell-mediated immune response during viral [80] and bacterial infections [79] and in a variety of diseases [81,82,83]. Neopterin level usually gradually increases in the beginning of any infections, indicating, so called, a productive infection. For COVID-19, it was found that neopterin had elevated concentrations (>10 nmol/L) within 9 days after the onset [84]. In the SERS spectra, the appearance of a band corresponding to neopterin provides information about the body’s early immune response to the SARS-CoV-2 virus attack. So far, there are only few reviews describing the significance of this immunomarker in the macrophage activation syndrome associated with SARS-CoV-2 infection [78]. The published data [85] indicate the importance of neopterin level, as it was found that infected patients with high level of neopterin helps to identifying patients at risk of a severe disease course [86]. Our spectral data indicate the formation and accretion of neopterin in the viral infected subject manifested by the appearance of the bands at 688 and 925 cm^−1^ mentioned above. To the best of our knowledge, for the first time, we recognized in the SERS spectra the contribution of vibration modes of this macrophage activation marker as the immunopathological response to SARS-CoV-2 infection. It is also crucial to understand the difference in neopterin levels between saliva and nasopharyngeal swabs. However, the immunopathology of COVID-19, including the role of neopterin, remains uncertain and further studies need to be conducted.

In order to analyze the mean intensity of the peaks between the two groups, the intensities of some bands were analyzed in relation to the intensity of the band at 1330 cm^−1^ (this band has the constant intensity for CoV(−) and CoV(+) averaged spectra). The calculated intensity ratio of the bands at 654 cm^−1^, 724 cm^−1^and 1445 cm^−1^ are shown in Appendix A. As can be seen, the relative spectral intensity of the bands at 724 cm^−1^ to 1455 cm^−1^ showed efficacy in the classification of the two types of samples: for CoV(+), the intensity ratio I_724_/I_1455_ equaled 1.03 ± 0.05, while for CoV(–) I_724_/I_1455_, it equaled 1.23 ± 0.04. All the observed SERS bands and their tentative assignments are presented in Table 1.

#### 2.2.1. Prediction by Means of PLS-DA Analysis

The calibration model was created from 51 CoV(+) and 53 CoV(−) nasopharyngeal swabs, while for validation purposes, eight CoV(+) and eight CoV(−) were exploited. For each sample, 15 single spectra were performed, giving 1560 spectra at the stage of calibration and 240 spectra at the stage of validation in total. The calibration model was based on the non-linear iterative partial least square (NIPALS) algorithm with random 10-segments cross validation. All the statistical parameters were comparable with the ones that were obtained for saliva calibration models (see Appendix A for comparison). The first thirteen variables that contributed the total of 67% of the variance in block Y and 83% of the spectral data (X matrix) of cumulative contribution fully describe the variability between analyzed data sets.

The score plots (Figure 2B) in 2D (above) show that two classes of CoV(−) and CoV(+) seemed to overlap significantly. However, 3D projections revealed that the data were grouped, forming CoV(−) and CoV(+) classes. The first latent variable (LV1) explained 27% of the variance in the block Y with 24% of the spectral data (X matrix), while the second latent variable (LV2) explained 10% of the Y matrix with 15% of original variance. Figure 2C demonstrates the weighted regression coefficient that was calculated for Factor-3, CoV(−) response. The variables at: 654, 749, 1004, 1046, 1210 cm^−1^, and 1454 cm^−1^ had a positive coefficient and 690, 728, 853, 925, 1760 cm^−1^ had a negative coefficient. The 1094 cm^−1^ variable with the weight of 0.03 and 690, 853, 925, and 1004 cm^−1^ with weights greater than 0.02 were the most crucial for calibration model.

The set of thirteen loading plots calculated for this association is presented in Appendix A. Factor-1, as the most influential, identified variables located at 689, 1445, 1582, and 1670 cm^−1^. The loading plot of Factor-2 revealed the importance of 1470 cm^−1^ and 1697 cm^−1^ and of F-3—1049, 1469, and 1700 cm^−1^ and the information was explained at the level of 8% and 7%. The band that were highlighted by the rest loading plots are presented on Appendix A.

The prediction analysis was performed in the same manner as for saliva samples. Most samples were correctly recognized at the stage on calibration with an accuracy of 98%, sensitivity of 100%, and specificity of 96%. The predicted values and standard deviations for samples taken for validation are listed in Appendix A. Most of the obtained predictive values were close to 1 or 0, with the standard deviation ranging between 0.17 and 0.33. Among these analyzed 16 samples, three CoV(+) and two CoV(−) samples had incorrect predictive values, giving the values of accuracy, sensitivity, and specificity as 69%, 63%, and 75%, respectively.

#### 2.2.2. Classification Results of PCA-LDA and SVMC Analysis

The original spectral data were compressed by principal component analysis (PCA) to 15 principal components and were then, in this form, utilized as input data for LDA calculations. The calculated discrimination plot for the CoV(+) and CoV(−) classes with the use of quadratic algorithm provided 89% accuracy at the level of a single spectrum (Appendix A). However, it is more reasonable to consider statistical parameters per patient, and in this respect, the accuracy, sensitivity, and specificity equaled 89%, 96%, and 83%, respectively. The obtained results indicate that two of 51 CoV(+) and ten of 53 CoV(−) samples did not completely match the typical spectral pattern of the majority of samples. The results of validation for 16 external samples with the particular attention to the number of spectra assigned to each class are presented in Appendix A. Since three out of eight samples stated as CoV(+) and two out of eight samples stated as CoV(−) were incorrectly classified, the accuracy, sensitivity, and specificity equaled 69%, 63%, and 75%, respectively (see Table 2).

In SVMC calculations, the optimal model was established with the Polynomial Kernel function (with degree equal four), 10-segments cross validation, and the parameters C = 0.01, γ = 0.1 giving all the spectra correctly recognized (training and validation accuracy 100%). From the validation results simulating the real diagnostic conditions, the SVMC provided an excellent sensitivity of 88% with respect to eight CoV(+) patients. In turn, 63% of specificity resulted from three of eight misclassified CoV(−) samples and was slightly lower than in the case of PCA-LDA. The accuracy calculated with respect to the total of 16 patients was 75%. All the detailed results of classification at the level of a single spectrum are presented in Appendix A and all the diagnostic parameters are set in Table 2 for comparison.

The above results demonstrate that nasopharyngeal swab is also a convenient clinical material that can reveal the biochemical changes resulted from COVID-19 and, therefore, enables differentiation between CoV(+) and CoV(−). The SVMC model offers the best diagnostic parameters at calibration stage among all methods and higher values of sensitivity and accuracy but the lower value of specificity at validation stage (63% vs. 75% for SVMC and PCA-LDA/PLS-DA, respectively). Comparing PLS-DA and PCA-DA, PLS-DA ensured a more accurate sample recognition at the calibration stage while maintaining the same diagnostic parameters at the validation stage.

## 3. Materials and Methods 

### 3.1. Viral RNA Extraction

The chemagic™ 360 automated extraction platform (PerkinElmer, Waltham, MA, USA) was used to extract SARS-CoV-2 RNAs from 300 µL of nasopharyngeal. Extraction was performed according to the manufacturer’s instructions named chemagic Viral DNA/RNA 300 Kit H96 (PerkinElmer, USA). Viral RNA was eluted with 80 µL elution buffer and used for RT-PCR assay.

### 3.2. SARS-CoV-2 RNA Detection Using Quantitative Reverse Transcriptase Real-Time Polymerase Chain Reaction (qRT-PCR)

The presence of SARS-CoV-2 was detected by qRT-PCR amplification of SARS-CoV-2 open reading frame 1ab (ORF1ab), nucleocapsid protein (NP) genes fragments, and a positive reference gene using DiaPlexQ™ Novel Coronavirus (2019-nCoV) Detection Kit (SolGent CO, Ltd., Daejeon, Republic of Korea). Conditions for amplifications were 50 °C for 15 min (reverse transcription), 95 °C for 15 min (initial PCR activation) followed by 45 cycles of 95 °C for 20 s (denaturation) and 60 °C for 40 s (annealing/extension). The result was considered valid only when the cycle threshold (Ct) value of the reference gene was ≤38. The result was considered positive when the Ct values of both target genes were ≤38, and negative when they were both >38. If only one of the target genes had a Ct value ≤38 and the other >38, it was interpreted as inconclusive.

### 3.3. SERS Platform Preparation

The SERS-active silicon substrates were prepared with a procedure described in detail in [50]. Briefly, the procedure of production SERS-active substrates involved three steps. In a first step, silicon wafer was cut into squares with the desirable dimension 3 × 3 mm by means of mechanical saw (the thicknesses of diamond grinding wheels 0.1 mm, the machining speed five mm/s). Second step was the physical modification of the silicon surface performed using femtosecond laser (λ = 1030 nm) working with the repetition rate of 300 kHz and pulse width of 300 femtoseconds. Then, to complete the preparation of the SERS substrate, in the last step, 100 nm layer of silver was sputtered using the PVD device (Quorum, Q150T ES, Laughton, UK). The resulting SERS-active platforms were uniformly covered with 100 nm silver layer, as can be seen on SEM image (Appendix A).

### 3.4. SERS Measurements

The measurements were performed using Bruker’s BRAVO spectrometer equipped with Duo LASER™ (700–1100 nm) and CCD camera. The laser power was 100 mW for both LASERs and the spectral resolution was 2–4 cm^−1^. Typically, 15 SERS spectra for each sample were acquired. Each spectrum was measured for 30 s. In these studies, we measured and presented the data based on 149 samples of saliva and 104 samples of nasopharyngeal swabs (15 spectra for each sample; total 3795 spectra). Both sets of samples (saliva and swabs) were from different individuals and were independent of each other. All experiments with clinical samples were conducted in accordance with relevant institutional regulations and guidelines and were approved by the Ethics and Bioethics Committee of Cardinal Stefan Wyszynski University in Warsaw.

### 3.5. Statistical Analysis

In the present work, the supervised and classification methods such as PLS-DA, PCA-LDA, and SVMC were applied for establishing classification models for CoV(+) and CoV(−) samples of saliva as well as nasopharyngeal swabs using the commercial Unscrambler^®^ software (CAMO software AS, version 10.3, Oslo, Norway). Before multivariate analysis, the SERS data were processed using the OPUS 7.2 software (Bruker Optic GmbH, 2012 version, Leipzig, Germany). For this purpose, the following manipulations were applied: smoothing (Savitzky Golay Filter: five points), baseline correction (concave rubberband correction; six iterations, six baseline points), cutting (in the range from 600 to 1700 cm^−1^), and normalization (Min-Max normalization). For more information about applied chemometric methods, see Appendix A.

## 4. Conclusions

In the present work, the numerical methods (PLS-DA, PCA-LDA, SVMC) were adopted to resolve and understand the biochemical changes in nasopharyngeal and saliva spectral responses resulting from SARS-CoV-2 infection and, therefore, enable differentiation between negative CoV(−) and positive CoV(+) groups of patients. The significant role of neopterin as the prospective biomarker in the prediction of COVID-19 infection from nasopharyngeal swab was indicated. In the case of SARS-CoV-2 infected saliva samples CoV(+), their biochemical composition was outlined by the increased contribution of methionine, nucleic acids of DNA/RNA and proteins such as ferritin, as well as specific immunoglobulins. Additionally, based on the multivariate methods and the calculated sensitivity, specificity, and accuracy, the saliva swabs sample was the more suitable, adequate, and reliable specimen for the SERS-based prediction of COVID-19 disease than the nasopharyngeal swabs sample.

The developed method is a promising alternative for rapid point of care diagnosis during the worldwide ongoing COVID-19 pandemic. However, it should be highlighted that in order to develop a reliable SERS-based diagnostic system for fast detection of COVID-19, a strictly defined methodology and protocol at each stage of the procedure including: preparation of clinical samples for SERS measurements (type of SERS-active support, time of sampling, collection, and processing of clinical subjects), defining of the spectral data acquisition (type and power of excitation laser, time of acquisition), and their exploration (chemometric analysis, machine learning, deep learning) should be applied and respected.

## Data Availability

Not applicable.

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
