# Peer review of "SERS Signature of SARS-CoV-2 in Saliva and Nasopharyngeal Swabs: Towards Perspective COVID-19 Point-of-Care Diagnostics"

_ijms, 2023, doi:10.3390/ijms24119706_

Round 1
Reviewer 1 Report
In this manuscript, Kamińska et al., demonstrate a sensitive and reliable SERS-based screening of COVID-19 disease using chemometric analysis. The approach is interesting and a simple and fast diagnosis of diseases is always a primary concern. Some points should be mentioned, prior to acceptance of your work for publication.
Recommendation – Minor Revision
1. All the Raman and SERS-based references attached are related only to the biomedical applications of the field. Reputed SERS and Raman papers can be added for the general introduction to SERS.
2. Authors can discuss the role of chemometric analysis in the introduction part.
3. The authors can include the SEM images of the substrate, and also discuss the reproducibility and RSD of the SERS substrates. How about the shelf-life of the SERS substrates?
4. Can the authors elaborate on the role of neopterin as a biomarker in predicting COVID-19 infection from nasopharyngeal swabs?
5. Tables can be shown in a better and neat way.
6. The authors can explain in detail how does the proposed analytical method address the challenges of specimen collection, analysis time, and screening sensitivity? A table can be included to compare the available and proposed methods in terms of sensitivity, cost, etc.
Minor corrections are needed.
Reviewer 2 Report
In this study Berus et al. used surface-enhanced Raman spectroscopy (SERS)-based approach coupled with chemometric analysis to report biochemical fingerprint of SARS-CoV-2 infected human saliva and nasopharyngeal swabs.
The study overall is very interesting and addresses a very important gap in our understanding of SARS-CoV-2 pathophysiology. The findings will be an important contribution to the relevant body of knowledge.
However, I have a few concerns and questions. My comments are as below:
1. Authors used human samples; however, ethical approval information is missing which should be added. Moreover, authors are advised to include a separate section in methodology (can be added in main text or SI) about patient/sample information. I recognize that a couple of sentences are included in results but that may be a bit confusing for readers. On page 3, lines 123-124, it is mentioned that saliva samples from a total of 149 human donors were used. Similarly, 104 swab samples were used (page 12, line 394). It is unclear whether saliva and swab samples were from the same patients or the two sets are mutually exclusive. Moreover, 20 and 16 saliva and swab samples, respectively, were used for external validation. Whether they are from different subject/patients, or there is a certain overlap between these two sets of samples? Also, why the number of samples is different at validation?
It is therefore requested that authors must provide all this information.
2. Why did the authors choose 1002 cm-1 band as reference? It is clearly evident from the Figure S1A that intensity of 1002 band is quite variable. While the authors have presented I654/I1002, I720/I1002, I1320/I1002, 163 I1445/I1002 ratios, the same information should also be included for the 1094, 1325, 1372, 1452, and 1690 bands.
Minor comment:
Authors have at some places included statements without providing relevant references at the appropriate place. For example
1. Page 4, lines 166-170. Authors need to cite recent studies which prove COVID replication in salivary glands and mucosae, without which this phenomenon cannot be observed in saliva samples. Ref 47 only explains the role of NP16, not replication of virus in the above mentioned tissues
2. Page 10, lines 352-354
Authors are requested to carefully review the whole manuscript to include relevant references at appropriate places
The manuscript would benefit from extensive language editing. There are numerous language issues throughout the text. The issues include: spelling/typing mistakes, poorly structured sentences that fail to communicate the intended message, as well as common grammatical errors. It is difficult to list all of the issues one by one here, therefore, I will present a few as examples. Authors are requested to improve write up in the light of these examples.
1. Page 3 lines 126-128
Poorly structured sentence. Hard for reader to grasp the authors' intended meaning. Typing mistake (averages instead of averaged)
2. Page 8, lines 276-278
"in the step of calibration" and "20 external ones" are examples of poor choice of words, which may not be appropriate for scientific write up
3. Page 12 line 384 "further studies need to be explored" studies are not explored, they are conducted.
4. Page 13 line 437 "three from eight" is wrong, it should rather be "three out of eight"
Reviewer 3 Report
In present MS authors describe use surface-enhanced Raman spectroscopy (SERS) and chemometric approaches for fingerprinting of SARS-CoV-2 infections. The authors developed the reliable strategies for rapid identification and differentiation of negative CoV(-) and positive CoV(+) samples with high accuracy, sensitivity, and specificity using human saliva specimen swabs. Moreover, the study established neopterin as the biomarker of COVID-19 infection in swab samples.
The study presents a significant approach for biological sample analysis using RNA and protein as biomarkers. The report is timely and could be considered for publication after careful revision.
I have few comments/suggestions to improve the MS:
1. In abstract, please include statistical analysis (significance and error) values for quantification/detection of RNA/proteins.
2. Please keep space between units and numbers, for example, 2μL in line 119, change it to 2 μL.
3. Authors need to enrich scientific literature on application of SERS in RNA detection/quantification. For example, PMID: 24460907; PMID: 28301688; and PMID: 24631541 in these studies authors demonstrated SERS could be used for RNA quantification. Please revise literature.
4. Line 191: Change ‘can be assignment’ to ‘can be assigned’
5. Conclusions are too broad. Need to be shortened and provide future implications of this study.
English needs some attention (minor editing).
